# Early Life Stress Affects *Bdnf* Regulation: A Role for Exercise Interventions

**DOI:** 10.3390/ijms231911729

**Published:** 2022-10-03

**Authors:** Taylor S. Campbell, Katelyn M. Donoghue, Urmi Ghosh, Christina M. Nelson, Tania L. Roth

**Affiliations:** 1Department of Psychological & Brain Sciences, University of Delaware, Newark, DE 19702, USA; 2School of Molecular & Cellular Biology, University of Illinois at Urbana-Champaign, Urbana, IL 61801, USA

**Keywords:** early life stress, neurotrophins, *Bdnf*, epigenetics, aerobic exercise

## Abstract

Early life stress (ELS) encompasses exposure to aversive experiences during early development, such as neglect or maltreatment. Animal and human studies indicate that ELS has maladaptive effects on brain development, leaving individuals more vulnerable to developing behavioral and neuropsychiatric disorders later in life. This result occurs in part to disruptions in Brain derived neurotrophic factor (*Bdnf*) gene regulation, which plays a vital role in early neural programming and brain health in adulthood. A potential treatment mechanism to reverse the effects of ELS on *Bdnf* expression is aerobic exercise due to its neuroprotective properties and positive impact on *Bdnf* expression. Aerobic exercise opens the door to exciting and novel potential treatment strategies because it is a behavioral intervention readily and freely available to the public. In this review, we discuss the current literature investigating the use of exercise interventions in animal models of ELS to reverse or mitigate ELS-induced changes in *Bdnf* expression. We also encourage future studies to investigate sensitive periods of exercise exposure, as well as sufficient duration of exposure, on epigenetic and behavioral outcomes to help lead to standardized practices in the exercise intervention field.

## 1. Introduction

Over 30 years ago, Dr. David Barker presented the Barker Hypothesis, which stated that the perinatal environment sets off a chain reaction of neural programming that determines cognitive function and emotional health in adulthood [1]. In line with this idea, the perinatal period is marked by a degree of neural plasticity that is highly sensitive to environmental influences and not seen during any later period of life [2,3,4]. During this developmental period, a cascade of neural processes work in tandem to program the brain [4]. Negative experiences during early development, such as stress, can alter the epigenetic regulation of neurotrophins [5,6,7,8] and thereby increase an individual’s susceptibility to later development of neuropsychiatric and neurodegenerative disorders [6,7,8,9]. Epigenetics refers to the addition of molecules, such as methyl groups, to DNA strands that alter the way that DNA can be read and used in the body. Hence, the consequences of these early disruptions can become embedded in the DNA and stay with a person through their entire lifespan.

Neurotrophins regulate brain development during infancy and adolescence, and in adulthood, they facilitate synaptic plasticity and neuronal survival [6,10,11,12]. Perturbations during development often dysregulate neurotrophin expression, leading to brain-region-specific maladaptive changes in expression [13,14,15,16]. In contrast, aerobic exercise typically upregulates neurotrophin expression, and is thought to be neuroprotective [17,18,19,20]. Most notably, exercise has a neuroprotective effect on the brain by preventing neuronal cell death [21] and facilitating adaptive cellular processes, including synaptogenesis (formation of synapses) [21] and neurogenesis [22]. For example, rodent studies show that aerobic exercise promotes neurogenesis in the hippocampus, a brain region where neurogenesis is abundant throughout adulthood [22,23]. This is important when considering that neurogenesis is reduced in the hippocampus of rodents exposed to developmental stress, making exercise a potential treatment mechanism for the maladaptive biological effects of stress [24,25,26,27]. Consistent with findings in animal subjects, imaging studies in humans indicate that child maltreatment leads to a decrease in hippocampal volume [28,29,30], as well as decreased hippocampal activation during threat detection [31] and memory tasks [32].

In this review we discuss the literature on rodent models of early life stress (ELS) and exercise interventions. We pay special attention to the effect of these experiences on neurotrophin regulation, with the capacity of exercise to correct biological processes and aberrant behavior associated with early stress.

## 2. Modeling Early Life Stress and Exercise in Rodents

In humans, ELS encompasses many different experiences. These range from physical and emotional abuse and neglect to living in a war-torn country, experiencing extreme poverty, or the death of a caregiver. In the laboratory, researchers recapitulate similar experiences in animal models to investigate the biological consequences of stress during early development. Two common models used are the limited bedding and nesting (LBN) model and the maternal separation (MS) model. In the LBN model, a rodent mother is not provided with sufficient nesting material to properly care for the pups. This causes stress in the dam and elicits more aversive behaviors from her towards the pups, such as actively avoiding and rough handling them, and less frequent or fragmented maternal behaviors such as arched back nursing and hovering over the pups [33,34]. This model is used to study the consequences of disrupted infant–caregiver interactions during early life. In the MS models, pups are separated from the dam for varying periods of time during early development [33,35] to approximate the experience of caregiver neglect.

Rodent models of aerobic exercise either use involuntary treadmill running or voluntary wheel running paradigms. Currently, the timing and duration of exercise exposure as a treatment intervention has not been standardized (see Figure 1). In voluntary models, rodents have continuous free access to running wheels, usually via a window cut out in the cage that allows the subjects to freely migrate to an attached wheel. Involuntary exercise models employ a treadmill apparatus that keeps the subject’s feet on the treadmill, forcing the subject to locomote for a set duration of time and intensity. Additional studies have employed both voluntary and involuntary aerobic exercise to compare the effects of the two running models. One study found that voluntary running wheel exercise decreased immobility time during a forced swim test of adolescent rats exposed to maternal separation; however, this antidepressant-like effect was lost in the involuntary exercised subjects [36]. Studies of this nature question the efficacy of forced exercise models as they likely upregulate the subject’s stress response, leading to a diminished neuroprotective effect. For example, Ke and colleagues [37] employed an animal model of stroke to measure the ability of aerobic exercise to recover motor behavior function and increase brain-derived neurotrophic factor (BDNF) protein levels in the hippocampus. Their results showed that voluntary, but not involuntary, exercise improved motor behavior overtime and increased BDNF in the hippocampus. In contrast, rats in the involuntary exercise group showed increased corticosterone (CORT) levels as well as a decrease in BDNF levels in the hippocampus compared to the control group [37]. These results are further supported by a 2016 study that found involuntary exercise to be maladaptive for stroke recovery in animal models. Svensson and colleagues [38] showed that, in a model of ischemic stroke, involuntary exercise increased anxiety-like behaviors on the open-field test (OFT), increased neuron loss in the right hippocampus, and increased fecal CORT levels following the OFT. They also noted a positive correlation between CORT levels and neuron loss [38]. In a 2014 study by Uysal and colleagues [39], voluntary exercise led to a decrease in basal CORT levels compared to both sedentary and involuntary exercised rats that was also accompanied by an increase in locomotion on the OFT in voluntarily exercised rats. Results from this study also showed that female rats exposed to voluntary exercise had increased BDNF protein levels in the prefrontal cortex (PFC) compared to sedentary rats. In male rats, BDNF levels increased in both exercise groups compared to the sedentary rats; however, there was a significantly greater increase in BDNF levels in the PFC of voluntarily exercised rats compared to the involuntary exercise group [39]. Taken together, these studies indicate that voluntary exercise models may be more advantageous when investigating the anxiolytic and neuroprotective effects of exercise, as involuntary exercise may exacerbate neural insults, including those caused by ELS, by upregulating CORT reactivity in the brain.

In addition to the type of exercise, the duration of exercise and age of exposure may play a role in its effectiveness. For example, Greenwood and colleagues [40] showed that six, but not three, weeks of voluntary wheel running was sufficient to prevent learned helplessness behaviors when subjects were exposed to uncontrolled tail shocks later. This same research group also showed that chronic voluntary exercise exposure is more rewarding in rats compared to short-term exposure. Six weeks, but not two weeks, of voluntary exercise led to exercise-induced changes in gene expression and receptor activity in the mesolimbic dopamine pathway that were accompanied by preference for a chamber that was previously paired with wheel running exposure on a conditioned place preference task [41]. Six weeks of voluntary exercise is also sufficient to decrease habituation time to future stress (loud noise exposure) as measured by significantly reduced plasma CORT levels compared to sedentary rats [42]. A breakdown of the ELS models and exercise intervention methods used in the current literature is provided in Table 1.

## 3. Behavioral Outcomes in Stress and Exercise Models

In laboratory models, the effects of ELS combined with later exercise experience vary based on the specific experimental parameters (see Table 2). Though nuances are present in the literature, a frequent finding suggests that rats with a history of ELS show increased anxiety- [52,53,58,60,62] and depressive-like [6,53,54,56,57,58,62] phenotypes, and these phenotypes are ameliorated by exercise exposure [36,51,52,53,54,56,57,58]. These outcomes are mostly illustrated in male rats exposed to MS, with information on female rodent outcomes and sex differences, as well as rats exposed to other models of early stress, severely lacking. However, there is some evidence that stress and exercise differentially affect sexes. For example, James and colleagues [52] showed that exercise ameliorates anxiety-like behaviors in male rats exposed to MS but worsens these behaviors in females exposed to MS. An outcome such as this underscores the importance of studying sex differences, including the effect of hormones on behavior and susceptibility to stress. Indeed recent studies have shown that estrogen can significantly impact the effect of trauma on the brain and susceptibility to psychiatric disorders [63,64]. 

The effects of exercise are not always consistent on ameliorating ELS-phenotypes, with increases in anxiety behavior [49,52] or no effects [36] sometimes observed. Rats may use the wheels to facilitate an escape behavior [44], which could have consequences for anxiety behavior. Another factor contributing to the inconsistency in exercise effects reflects the use of a voluntary WR treatment verses an involuntary treadmill (TM) exercise treatment. For example, Sadeghi and colleagues [36] reported that WR exposure decreased depressive-like behavior in rats but that TM exposure had no effect. Outside of affecting any ELS outcomes, exercise did bolster behavioral performance on cognitive and memory tasks in several studies [49,61], as is a common finding in the exercise literature [65].

## 4. Epigenetics and Neurotrophins

One way our experiences can get under the skin to affect genes, including the *Bdnf* gene, is through epigenetic mechanisms. *Epigenetics* refers to modifications to DNA that affect gene expression without making changes to the genetic sequence. One form of epigenetic regulation is called DNA methylation, wherein a methyl group is added to the cytosine at a CG site (cytosine-guanine dinucleotide) on the DNA [66,67]. CG sites are highly potent surrounding the promotor regions of most genes, making them a prime target for gene regulation. Increased methylation at promoter regions typically leads to decreases in gene expression because methyl groups recruit repressor proteins, interact with chromatic structure, and inhibit transcription factors from binding [66,68]. 

As researchers look to understand how early life stress can have long-term behavioral consequences and how exercise can reprogram the brain to have neurotherapeutic effects, focus often turns to neurotrophins, especially BDNF (see Figure 2 for our theoretical framework). Neurotrophins are a family of proteins which induce the development, survival, and function of neurons [6,69,70]. BDNF’s neurotrophic actions are vital for brain development and plasticity, and BDNF exhibits activity-regulated release in the central nervous system [71,72,73]. BDNF is a neurotrophin important for neural development, neural plasticity, learning, memory, and synaptic plasticity later in life, especially within the hippocampus [74,75]. Typically, increased methylation of the *Bdnf* gene is associated with decreased expression of its genetic material [68]. Methylation at any of *Bdnf’s* nine promotor regions can lead to decreased transcription of total *Bdnf* mRNA [76]. Stress during neonatal development has the capacity to alter *Bdnf* methylation for the long haul [5,77,78], which is important as decreased BDNF protein expression is found in patients with neurogenerative diseases and neuropsychiatric disorders [79,80,81,82]. These data highlight *Bdnf* as an important genetic locus for studies investigating epigenetic-behavioral interactions.

## 5. Stress, Exercise and Neurotrophins

Many studies have shown that ELS reduces levels of both *Bdnf* mRNA and BDNF protein in multiple brain regions, including the prefrontal cortex and hippocampus [5,70,77,78,83,84,85]. Early life exposure to stress, especially within a caregiving environment, can result in a decrease in *Bdnf* gene expression through increased methylation of the *Bdnf* gene [5,86,87,88]. This impact of developmental stress extends to humans. For example, *Bdnf* DNA methylation correlates with the number of aversive childhood experiences in patients with bipolar disorder [89]. Further research in humans sheds light on the transgenerational effects of ELS on BDNF, in that babies born to mothers who experienced ELS show changes in *Bdnf* methylation and expression in blood cells obtained from the umbilical cord based on infant sex and degree of maternal fear [90]. 

While ELS generally decreases neurotrophin levels, aerobic exercise increases neurotrophin expression and is thought to be neuroprotective [17,91,92,93]. Exercise has positive impacts on neurotrophin expression, which directly impact neuronal survival and neurogenesis. Previous studies have identified exercise as a behavioral mechanism that specifically increases *Bdnf* expression [55,56,93] and decreases *Bdnf* methylation [94,95]. Several studies have also reported significant associations between exercise-induce BDNF upregulation and improved cognition [96] and depression symptoms [97].

The effect of exercise on neurotrophin expression in rodents exposed to ELS is understudied, with only 6 studies beginning to elucidate this relationship to date. Within these studies, exercise affected *Bdnf*/BDNF expression in a nonuniform and nuanced manner. Three studies showed that voluntary WR increased *Bdnf* mRNA in the hippocampus following 3hrs/daily MS during the first 2-3 weeks of life [48,55,59], and this increase was associated with rescued hippocampal neurogenesis in the dentate gyrus [48]. At the protein level, one study found that exercise increased BDNF expression in the striatum but not the ventral hippocampus compared to sedentary MS-exposed rats [56]. Given the complicated nature of *Bdnf* expression and gene regulation, Wearick-Silva and colleagues [61] investigated the exon-specific effects of exercise in the hippocampus. They reported that MS decreased *Bdnf* exon IV expression and increased exon IX expression, while exercise had an opposite effect on exon IX and increased *Bdnf* exon I expression. To shed light on the mechanism behind exercise-induced increases in *Bdnf*/BDNF expression, future studies should measure *Bdnf* exon-specific methylation in conjunction with expression to determine if specific genetic loci act in tandem to alter de novo expression. 

Little is known regarding the effect of exercise on other neurotrophins in this ELS context. Marais and colleagues [56] produced the only current study investigating NT-3 and nerve growth factor (NGF) in this model, where they reported no significant effects on these neurotrophins in the ventral hippocampus and striatum. However, further investigation is warranted given that areas known to be highly impacted by stress and exercise, including the cerebellum, PFC, and dorsal hippocampus [98,99], have been overlooked. 

## 6. Call to Action: Further Exploring ELS & Exercise

ELS alters long-term neurotrophin expression in the brain. These epigenetic changes contribute to an individual’s risk of numerous neurological, immune, and psychiatric disorders. We propose use of aerobic exercise as a treatment mechanism in future studies to understand the capacity of exercise to bolster the brain and body against ELS-induced disruption of biological processes. Support for our line of thinking comes from studies showing that exercise improves spatial memory, autoimmune and neurodegenerative disease symptomatology, muscle function, and gut health with concomitant changes in various neurotrophin expression levels. 

Exercise has become a popular research area in neuroprotective research fields for its promising effects on brain health. A 2017 study following the natural aging of older adults (for ~10 years) showed that exercise is positively related to total cerebral and hippocampal volumes, and negatively related to developing Alzheimer’s Disease and dementia [100]. This suggests exercise is neural protective against neuronal and glial cell loss through the lifespan. It’s widely accepted that exercise has a positive effect on BDNF expression and epigenetic regulation [94,101], however, the data on other neurotrophins is much more nuanced and understudied [102]. Currently, only two studies have investigated the effect of exercise on NT-3 expression in humans [103,104]. A 2021 cutting-edge study reported that 12 weeks of high-intensity interval training increased serum levels of BDNF, NGF, NT-3, and NT-4 in elderly, obese, Chinese subjects [104]. Importantly, future research must critically examine exercise and participant parameters as these variables seem to be important when comparing participant outcomes. For example, in a 2018 clinical trial of adult obese males, high-intensity interval training (HIIT) had no effect on blood BDNF, NT-3, or NT-4 [103]. However, resistance training increased NT-3 and NT-4 levels, and combined exercise (resistance training plus HIIT) increased NT-3 and BDNF levels. In the combined exercise group, BDNF and NT-3 levels were positively correlated [103]. Taken together, these studies underscore the importance of continued research at the intersection of exercise science and neuroscience. 

Further research indicates that aerobic exercise may be a valuable treatment mechanism for neurodevelopmental, autoimmune and psychological disorders. Aerobic exercise is a popular intervention strategy in studies investigating the molecular and cognitive effects of fetal alcohol spectrum disorders, with exercise exposure ameliorating *Bdnf* dysregulation [94], corpus callosum volume deficits [105], and executive functioning [106]. Recent work also demonstrates that aerobic exercise decreases the pathogenesis of multiple sclerosis [107], while increasing peripherally circulating BDNF and NGF [108]. Moderate physical exercise for 6 weeks is sufficient to reduce self-reported depression levels and increase peripheral BDNF and NGF levels in postmenopausal woman [109]. The therapeutic benefits of exercise on disease states underline a promising future for exercise intervention models.

## Figures and Tables

**Figure 1 ijms-23-11729-f001:**
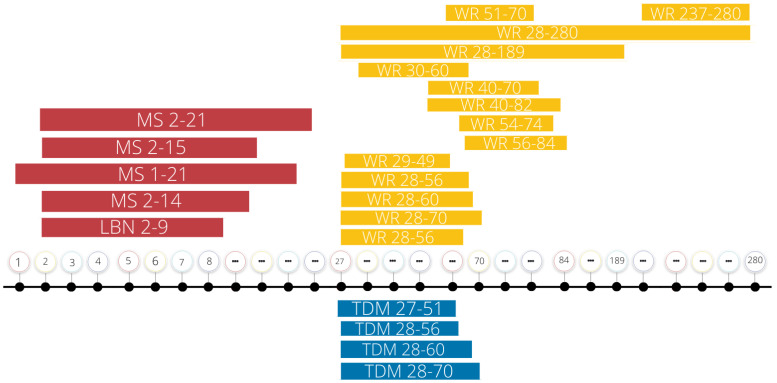
Summary of current ELS and exercise intervention protocols. This summary figure represents the days, recorded in postnatal days, of early life stress exposure and exercise interventions recounted in this review. Note the variability in experimental parameters. MS, maternal separation; LBN, limited bedding and nesting; WR, wheel running; TDM, treadmill; “…”, breaks in the timeline.

**Figure 2 ijms-23-11729-f002:**
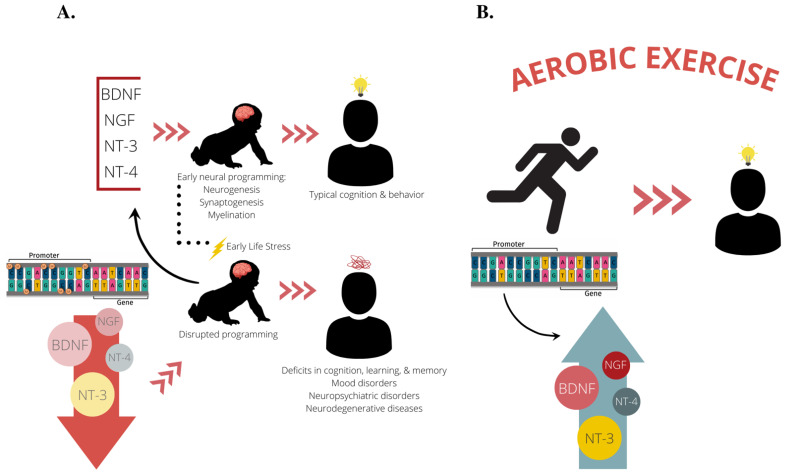
Our proposed theoretical framework. (**A**) Based on previous studies, it is known that BDNF, NGF, NT-3, and NT-4 facilitate early neural programming of the brain by promoting neuronal survival and outgrowth during the perinatal period. Proper neural development during early life facilitates cognitive abilities and protects against neurobehavioral insults later in life, such as depression and anxiety disorders and neurodegenerative diseases. Experiencing early life stress can lead to dysregulation of these vital neurotrophins through epigenetic mechanisms such as DNA methylation, often leading a reduction in neurotrophin expression. This leads to disruptions in early neural programming that can stay with an individual throughout their lifespan and put them at an increased risk for developing deficits in cognition, learning, and memory, as well as neuropsychiatric disorders and neurodegenerative diseases. (**B**) We propose aerobic exercise as a treatment mechanism to normalize neurotrophin expression and bolster cognition and neural health by reversing epigenetic perturbations set forth in early life. BDNF, brain-derived neurotrophic factor; NGF, nerve growth factor; NT-3, neurotrophin-3; NT-4, neurotrophin-4.

**Table 1 ijms-23-11729-t001:** Summary of Methods in Currently Published ELS and Exercise Intervention Studies.

Subjects	Sex	Form of ELS	Form of Exercise	Reference
C57BI/6J Mice	Both	LBN PN2-9	Access to running wheel (voluntary exercise) PN237-280	[43]
Sprague Dawley Rats	Male	MS PN2-14; 3 h daily	Access to running wheel (voluntary exercise) PN29-49	[44]
Sprague Dawley Rats	Male	MS PN2-14; 3 h daily	Access to running wheel for 5 days a week PN40-82 (voluntary exercise, pair-housed)	[27]
Sprague Dawley Rats	Male	MS PN2-14; 3 h daily	Access to running wheel PN54-74 (voluntary exercise)	[45]
C57B1/6 Mice	Male	MS PN1-21; 3 h daily	Access to running wheel during 4–27 weeks of age (voluntary exercise, pair-housed)	[46]
C57B1/6 Mice	Female	MS PN1-21; 3 h daily	Access to running wheel (voluntary exercise) at either 4 (pair-housed) or 8 (single-housed) weeks of age	[47]
C57B1/6 Mice	Male	MS PN1-21; 3 h daily	Access to running wheels 4–8 weeks of age (voluntary exercise, pair housed)	[48]
Sprague Dawley Rats	Male	MS PN2-14; 3 h daily	Access to running wheel (voluntary exercise) PN29-49	[49]
Sprague Dawley Rats	Male	MS PN2-14; 3 h daily	Access to running wheels PN54-74 (voluntary exercise)	[50]
Sprague Dawley Rats	Male	MD PN2-14; 1 h daily	Running on a treadmill for 6 weeks beginning at PN28 for 10 min/day for the first 5 days and incrementally increased to 60 min/day at a speed of 9 m/min (week 1), 12 m/in (week 3), and 15 m/min (week 6) (involuntary exercise)	[51]
Wistar Rats	Both	MS PN2-14; 3 h daily	Access to running wheels for 1 h per day, 5 days a week during PN40-70 (voluntary exercise)	[52]
Albino Wistar Rats	Male	MS PN2-14; 3 h daily	Access to running wheels PN28-60 (voluntary exercise, pair housed)	[53]
Sprague Dawley Rats	Male	MS PN2-14; 15 or 180 min/day	Access to running wheel (voluntary exercise) PN51-70	[54]
Sprague Dawley Rats	Male	MS PN2-14; 3 h daily	Access to running wheel (voluntary exercise) during weeks 4–10 of age (3–4 animals per cage)	[55]
Sprague Dawley Rats	Male	MS PN2-14; 3 h daily	Access to individual running wheels 5 days per week PND40-82 during dark cycle only (voluntary exercise; pair housed with perforated plexiglass divider during WR access).	[56]
Albino Wistar Rats	Male	MS PN2-14; 3 h daily	Access to wheel running on PN21 then continuous access to running wheel PN28-60 (voluntary exercise)	[57]
Sprague Dawley Rats	Male	MS PN2-21; 3 h daily	Running on a treadmill for 4 weeks, 6 days/week; 3 m/min 5 min warm-up; 10 m/min 30 min exercise (first 2 weeks) or 12 m/min 40 min exercise (last 2 weeks), and 3 m/min 5 min cool-down (involuntary exercise)	[58]
C57B1/6 Mice	Female	MS PN1-21; 3 h daily	Access to running wheels at 4–8 weeks of age(voluntary exercise, pair-housed)	[59]
Albino Wistar Rats	Both	MS PN2-14; 3 h daily	2 conditions PN28: Access to running wheel PN28-60 (voluntary exercise) OR treadmill running 5 days/week from PN28-60 for 30 min at 10 m/min (first 2 weeks), 45 min at 15 m/min (week 3), and 60 min at 15 m/min (week 4) (involuntary exercise)	[36]
Albino Wistar Rats	Male	MS PN2-14; 3 h daily	Access to running wheels PN28-60 (voluntary exercise)	[60]
Balb/c mice	Female	MS PN2-15; 180 min daily	Running on a treadmill 60 min/day at 10 m/min, 5 days per week from PN27-51 (involuntary exercise)	[61]
Rats (not specified)	Male	MS PN2-14; 3 h daily	2 conditions PN28: Access to running wheel for 4 weeks (voluntary exercise) OR treadmill running 5 days/week for 4 weeks for 30 min at 10 m/min (first 2 weeks), 45 min at 15 m/min (week 3), and 60 min at 15 m/min (week 4) (involuntary exercise)	[62]

LBN, limited bedding and nesting; MS, maternal separation; PN, postnatal day.

**Table 2 ijms-23-11729-t002:** Summary of Behavioral Findings in Currently Published ELS-Exercise Intervention Studies.

Subjects	Sex	Form of Early Life Stress	Age of Behavioral Test	Behavioral Outcome	Exercise Type	Reference
NMS Mice	Female	MS PN1-31, 3 h daily	PN56	MS increased light sensitivity in the LDB (migraine mouse model); outcome ameliorated by WR	WR	[47]
Sprague-Dawley Rats	Male	MS PN2-14, 3 h daily	PN29	WR exposure increased anxiety (EMP, OFT) WR exposure improved temporal memory (TO) and spatial learning (MWM) No effect on OIP, NOR, or OLT and no effect of MS	WR	[49]
Sprague-Dawley Rats	Male	MD PN12-25, 1 h daily	PN28	MD increased immobility time on the FST and time spent in the light versus dark (LDB) WR normalized/decreased immobility time (FST) and time spent in the light box (LDB)	TM	[51]
Wistar Rats	Both	MS PN2-14, 3 h daily	PN75-79	MS decreased total OFT locomotion in males and females. WR normalized locomotion in males but further decreased locomotion in females No effect on EPM open arm entries	WR	[52]
Albino Wistar Rats	Male	MS PN2-14, 3 h daily	PN21	MS decreased open arm entries and time spent in open arms on the EMP and increased immobility on the FST; WR normalized behavior	WR	[53]
Sprague-Dawley Rats	Male	MS PN2-21, 5 h daily	PN28; 75–81	MS decreased center duration and entries on the OFT (PN28; pre-WR) No effect of OFT locomotion or center time/entries in adulthood MS decreased open arm entries and time spent in open arms on the EMP; WR normalized behavior No effect of NOR, OIP, or TO	WR	[54]
Sprague-Dawley Rats	Male	MS PN2-14, 3 h daily	PN40	MS increased FST immobility; WR normalized immobility	WR	[56]
Albino Wistar Rats	Male	MS PN2-14, 3 h daily	PN60	MS increased FST immobility and decreased grooming during SPL; WR normalized behavior No effect of OFT	WR	[57]
Sprague-Dawley Rats	Male	MS PN2-21, 3 h daily	PN22	MS decreased open arm entries on the EPM, increased FST immobility, and decreased OFT center locomotion WR increased open arm entries and time spent in open arms (EPM), decreased FST immobility, and increased OFT center locomotion	TM	[58]
Albino Wistar Rats	Both	MS PN2-14, 3 h daily	PN21	MS increased immobility on the FST, decreased sucrose preference (SPT), and decreased grooming during splash test; WR (but not TM) normalized behavioral outcomes No effect on OFT	WR/TM	[36]
Albino Wistar Rats	Male	MS PN2-14, 3 h daily	PN60	MS increased immobility on the FST, decreased sucrose preference (SPT) and decreased grooming during the SPL WR decreased immobility (FST) and normalized grooming time (SPL) No effect on OFT	WR	[60]
Balb/c Mice	Female	MS PN2-15, 3 h daily	PN24	MS impaired NOR performance TM normalized NOR performance	TM	[61]
Rats (not specified)	Male	MS PN2-14, 3 h daily	PN61-70	MS decreased center time on the OFT and arm time and arm entries on the EPM; WR or TM normalized these behaviors MS increased immobility on the FST and decreased sucrose consumption (SPT); WR but not TM normalized these behaviors	WR/TM	[62]

MS, maternal separation; WR, wheel running; TM, treadmill running; LDB, light/dark box; EMP, elevated plus maze; OFT, open-field test; TO, temporal order task; MWM, Morris water maze; OIP, object in place; NOR, novel object recognition; OLT, object location task; SPL, splash test; SPT, sucrose preference test.

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
