# Peer review of "Early Life Stress Affects Bdnf Regulation: A Role for Exercise Interventions"

_ijms, 2022, doi:10.3390/ijms231911729_

Round 1

Reviewer 1 Report

The review entitled “Early life stress affects neurotrophin regulation: A role for exercise interventions” by Campbell et al. is interesting, comprehensive, and well written. I suggest only a few minor revisions:

Fig. 1. The legend should explain some points, such as the meaning of color shades and different colored points in the time scale.

Table 2. In the title, funding should be replaced by findings

Reference 27. The citation format has to be revised.

Reviewer 2 Report

The paper by Dr Campbell et al., entitled "Early life stress affects neurotrophin regulation: A role for exercise interventions", is an interesting review dealing with the deleterious effects of early life stress (ELS) on brain development, with a focus on the ELS-induced alterations in neurotrophin expression during the early life.

The review is well organized and it addresses the main focus with clarity and appropriate references. My only suggestion is to re-title the review, since it does not address the effects of ELS on all neurotrophins but BDNF. Abstract should be re-written accordingly.
